# Ovine Bone Morphology and Deformation Analysis Using Synchrotron X-ray Imaging and Scattering

**Eugene S. Statnik** [1,2,*], **Alexey I. Salimon** [2,3], **Cyril Besnard** [4], **Jingwei Chen** [4], **Zifan Wang** [4], **Thomas Moxham** [4,5], **Igor P. Dolbnya** [5] **and Alexander M. Korsunsky** [3,4,5,*]

1. Centre for Design, Manufacturing and Materials, Skoltech, 121205 Moscow, Russia
2. Centre of Composite Materials, National University of Science and Technology MISiS, 119049 Moscow, Russia; A.Salimon@skoltech.ru
3. HSM lab, Centre for Energy Science and Technology, Skoltech, 121205 Moscow, Russia
4. MBLEM, Department of Engineering Science, University of Oxford, Parks Road, Oxford OX1 3PJ, UK; cyril.besnard@eng.ox.ac.uk (C.B.); jingwei.chen@linacre.ox.ac.uk (J.C.); zifan.wang@exeter.ox.ac.uk (Z.W.); thomas.moxham@eng.ox.ac.uk (T.M.)
5. Diamond Light Source, Harwell Science and Innovation Campus, Didcot, Oxfordshire OX11 0DE, UK; igor.dolbnya@diamond.ac.uk
* Correspondence: Eugene.Statnik@skoltech.ru (E.S.S.); alexander.korsunsky@eng.ox.ac.uk (A.M.K.)

**Abstract:** Bone is a natural hierarchical composite tissue incorporating hard mineral nano-crystals of hydroxyapatite (HAp) and organic binding material containing elastic collagen fibers. In the study, we investigated the structure and deformation of ovine bone by the combination of high-energy synchrotron X-ray tomographic imaging and scattering. X-ray experiments were performed prior to and under three-point bending loading by using a specially developed in situ load cell constructed from aluminium alloy frame, fast-drying epoxy resin for sample fixation, and a titanium bolt for contact loading. Firstly, multiple radiographic projection images were acquired and tomographic reconstruction was performed using SAVU software, following segmentation using *Avizo*. Secondly, Wide Angle X-ray Scattering (WAXS) and Small Angle X-ray Scattering (SAXS) 2D scattering patterns were collected from HAp and collagen. Both sample shape and deformation affect the observed scattering. Novel combined tomographic and diffraction analysis presented below paves the way for advanced characterization of complex shape samples using the Dual Imaging and Diffraction (DIAD) paradigm.

**Keywords:** residual stress/strain; ovine bone; synchrotron X-ray diffraction; *SAVU* tomography reconstruction; *Avizo* segmentation

## 1. Introduction

Skeletal bones form the crucial structural framework of mammalian skeleton that sustains mechanical loading in the course of movement, daily work, sport, exertion, and trauma. Its safe performance is crucial for the functioning of human body, but can be affected and compromised by arthritis, osteoporosis, cancer, and many other diseases. According to the International Osteoporosis Foundation (IOF), osteoporosis alone is responsible to an estimated 75 million fractures annually according to the statistics from Europe, USA, and Japan [1].

The study of the structure and function of bones has been a focus of attention for researchers in microscopy and anatomy ever since the application of microscopy to the study of natural tissues began, notably as reported by Hooke in his treatise *Micrographia* [2]. The interest in the structure and performances of bones has continued to increase, raising scientific interest and motivating numerous

research studies that make use of the most advanced characterization techniques allowing the neat hierarchy of structural elements to be revealed at different scales [3] as following:

(a)　Macrostructure (cortical and trabecular bone tissues, of extent of many mm);
(b)　Meso-structure (osteons, 500–1 mm);
(c)　Microstructure (Haversian canals, 10–500 μm);
(d)　Sub-microstructure (lamellae, 1–10 μm);
(e)　Nanostructure (collagen fibrils, less than 1 μm).

An important contribution to the current understanding of bone mechanics has been made by the German anatomist and surgeon Wolff (1836–1902) who stated that a healthy bone adapts to the mechanical loads it experiences, a loaded bone will gradually remodel itself to become stronger, as can be noticed in the internal architecture of the trabecular and cortical bone parts. The large scale (mm range) changes of this kind can be revealed by conventional X-ray radiography and tomography; however, microscale processes responsible for bone adaptation require further detailed analysis. Importantly, the inverse process also occurs, that is, in the absence of loading, a bone tends to become less dense and weaker, having important implications for disease treatment and replacement surgery. *Mechanotransduction* is the term that describes bone remodelling in response to loading. It involves the conversion of mechanical parameters (stresses and strains) converted into biochemical signals through electromechanical coupling, biochemical signal generation and transmission, and cell response. It is in this context that the *mechanostat* hypothesis (or theorem) has been introduced [4]. The complexity of the bone remodelling process lies in the dependence of bone structure evolution on the magnitude duration, and rate of loading, and the significant importance of cyclic load on bone formation. The deduction of firm insights into the causal relationships in bone remodelling is complicated by the need to conduct observations in vivo. Nevertheless, the accumulated laboratory and clinical evidence [5] supports the applicability of this view.

In an extension of the above set of views, we consider the bone as an internally pre-stressed composite consisting of hard and somewhat brittle hydroxyapatite (HAp) mineral nanocrystals held together by the organic matrix containing a network of collagen fibers that possess high tensile strength. The intricate stress balance between collagenous network and HAp crystallite "filling" is closely connected with the process of bone remodelling, particularly in view of the piezoelectric response of collagen that provides a conversion mechanism between deformation and stress, on the one hand, and electric signals within the tissue. The complexity of elucidating the precise micro-mechanisms that underlie the *mechanotransduction* phenomenon is their fine scale and transient nature, calling for advanced in situ and operando experiments being conducted using suitable non-destructive methods.

Prior research into hard mineralized human tissues (dental enamel and dentine, [6]) has shown the importance of internal stress balance between components of distinct composition and mechanical properties for the function of load-bearing organs. Residual stresses in bone are known to develop during growth and remodelling and play an important role in determining its strength [7].

In view of the considerations above, we put forward long ranging views of the principal mechanisms and analytical approaches that need to be addressed to elucidate the physiological function of the bone. We surmise that the interaction between the mineral and organic content of the bone maintains it in an internally stressed state. These internal stresses change in response to external loading, and guide bone structure evolution through the process of mechanotransduction. We refer to the concept of co-evolving internal stress state as Osteodynamic Equilibrium (*OsteodynE*) that involves the load transfer and accommodation of deformation within the organic and mineral parts of the bone and ensures that both compressive and tensile loads are sustained by the collagen-HAp composite in the optimal way.

Bone tissue is the subject of extensive research covering various aspects of its biology, morphology, pathology, growth, remodelling, and reconstructive surgery. Classical fracture mechanics analysis of bone tissue has only been pursued recently, and with it came the realization that the complexity of

crack morphology raises additional challenges in applying fracture mechanics methods. The limiting load-bearing capacity of bone that determines the onset of major damage, ultimately leading to fracture. It is also important to note that the process of bone micro-scale damage is also well-documented, but remains insufficiently understood due to the fine scale at which it unfolds. Efforts to apply fracture to bone tissue analysis have been made [8]; however, many aspects of crack generation and propagation mechanisms within the complex stress field are still not fully understood. Tubular bones with complex shape and microstructure are particularly susceptible to fracture under bending, making this particular mode of loading especially suitable for conceptual and experimental analysis, as it allows considering both compression and tension modes within one loading configuration.

Bone fracture mechanics concerns microcrack damage, crack initiation, and propagation [9]. The application of fracture mechanics principles to complex bone structure requires account to be taken of the presence of fibrous elastic collagen and mineral hydroxyapatite (HAp).

During the last two decades, a number of attempts were made to investigate stresses in the bone using a variety of techniques. Almer et al. [10] studied internal stresses within bone under in situ compressive loading. Tadano et al. contributed to residual stress/strain evaluation in bone tissues using high-energy synchrotron X-ray diffraction [11–15]. In contrast, Nazer et al. tried to obtain the strain of bone tissue by invasive procedures using strain gauges glued to the bone surface [16]. Other researchers tried to investigate residual stress by the nanoindentation technique. For instance, Rho et al. studied variations in the properties of individual thick osteon lamella by nanoindentation [14]. A number of reports concerned the relationship between bone tissue strain and lattice strain of HAp crystals in cortical bone under tensile loading [17,18] and the exploration of residual stresses in collagen fibrils within rings of arterial tissues [19] by Wide-Angle X-ray Scattering (WAXS) and Small-Angle X-ray Scattering (SAXS) techniques, respectively. However, many questions remain open that need to be addressed in the context of complex bone shapes and structural elements. Synchrotron X-ray tomography has been used to visualize both the structure and the fracture progress in bone [20].

From another viewpoint, in recent years it has become apparent that synchrotron X-ray diffraction can provide unique insights into the nature of strain fields in the vicinity of a crack tip, including complex structured materials such as bone [21–26]. Our recent studies have shown how this can be used to obtain insights into the effects of crack closure [27] that may be associated with fatigue crack overloads and underloads. Further understanding of the complex response of bone to fracture is possible by combining experimental investigations with advanced numerical modelling, which requires validation at appropriate scales. For example, cancellous bone is a prominent example of a multi-scale porous material, whereas the cortical bone tissue is intrinsically a composite of fibrous collagen matrix highly filled with HAp crystallites, with a hierarchy of pores distributed in a semi-ordered manner.

In this article, we report the results of a series of experiments to combine tomography and diffraction that aim to fill the gap in the existing knowledge of the structure-deformation-strength relationship in bone. We describe the newly developed protocol for preparing complex-shaped bone samples for in situ loading. For the purpose of analysis, tomographic reconstruction and segmentation algorithms were applied, followed by the recently published technique for fast interpretation of 2D X-ray diffraction patterns [28] to evaluate the deformation state in an ovine (lamb) bone.

## 2. Materials and Methods

### 2.1. Sample Preparation

A set of fresh (max 6 h from abattoir) ovine rib bones were purchased at Oxford Covered Market and soaked in formalin fixing solution for 1 day. Rib bones were cleaned from soft tissue, tendons and ligaments using surgical scalpels. Clean bones were placed back into formalin solution for further kept 8 to 36 h. Each sample was surface-dried with paper towel for 5 min, and the ends were cemented into 3D-printed hollow grip cylinder forms using fast epoxy resin (2 h setting time to maximum stiffness).

The finished specimens are shown in Figure 1. Particular attention was paid to preserving coaxial alignment of the grip cylinders.

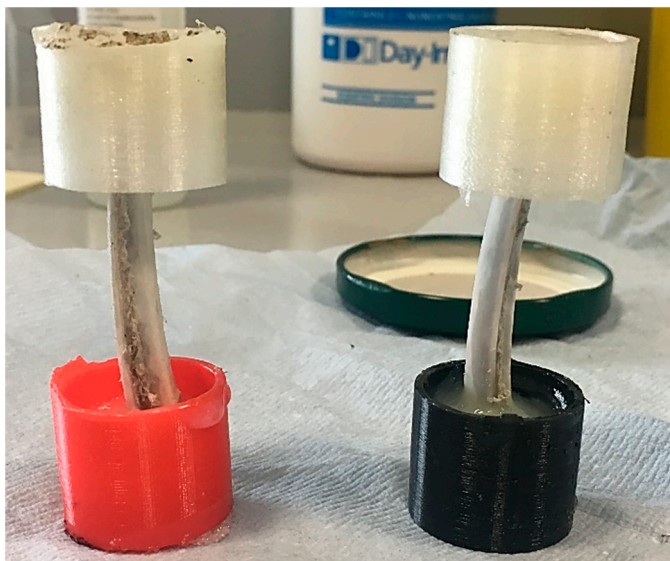

**Figure 1.** Finished samples of ovine bone for in situ loading.

*2.2. Synchrotron X-ray Multiple Projection Imaging for Tomography Analysis*

Prior to conducting in situ 3-point bending experiments of bone samples, tomographic datasets were collected from each sample to allow precise determination of their exterior shape and internal structure. To this end, a $6 \times 12$ mm$^2$ box X-ray beam was used to illuminate the full horizontal width of the sample(s) during rotation through 180°. The beam was attenuated (0.5 mm of Al + 0.5 mm of Cu) and monochromated to ~18 keV, and radiographic images were acquired using the PCO.edge X-ray "white" beam compatible camera system [29] tuned to the effective resolution of ~5 μm The optimal acquisition time was set to 0.1 s. The number of flat field and dark field images acquired was 20. Each image was normalized according to standard procedures used in X-ray imaging.

*2.3. Synchrotron X-ray Scattering*

High-energy synchrotron X-ray imaging and diffraction (SAXS and WAXS) experiments were carried out at the B16 Test beamline at Diamond Light Source (Harwell, Oxford, UK) [30]. The photon energy of 20 keV was selected using the Ru/B4C double-multilayer monochromator (DMM), and the beam energy never changed during the experimental session. For diffraction experiments, the beam was collimated down to $100 \times 100$ μm$^2$ using several pairs of slits to obtain a well-defined beam with controlled divergence to enable SAXS data collection. Exposure times of 5 and 30 s were used for WAXS and SAXS pattern collection, respectively. 2D WAXS patterns were acquired using Image Star 9000 detector (Photonic Science Ltd., Saint Leonards, UK) with a $3056 \times 3056$ pixels area ($95 \times 95$ mm$^2$, 16-bit dynamic range). The sample-to-detector distance of ~160 mm was calibrated using a lanthanum hexaboride (LaB$_6$) sample. SAXS patterns were recorded using a Pilatus 300K 2D detector (Dectris, Baden-Daettwil, Switzerland) with a $487 \times 619$ pixels area (84(h) $\times$ 106(v) mm$^2$, 20-bit dynamic range). The SAXS detector was positioned at ~6 m from the sample, with the sample-to-detector distance calibrated using a standard dried collagen sample (lyophilized rat tail). Typical WAXS and SAXS patterns are shown in Figure 2. The beam was first defind by slits S3 acting as a virtual source. S4 slits were used as pre-sample guard slits to cut off parasitic scattering background from S3 slits. The combination of slits set the beam size at $0.1 \times 0.1$ mm for WAXS and $0.05 \times 0.05$ mm for SAXS, respectively.

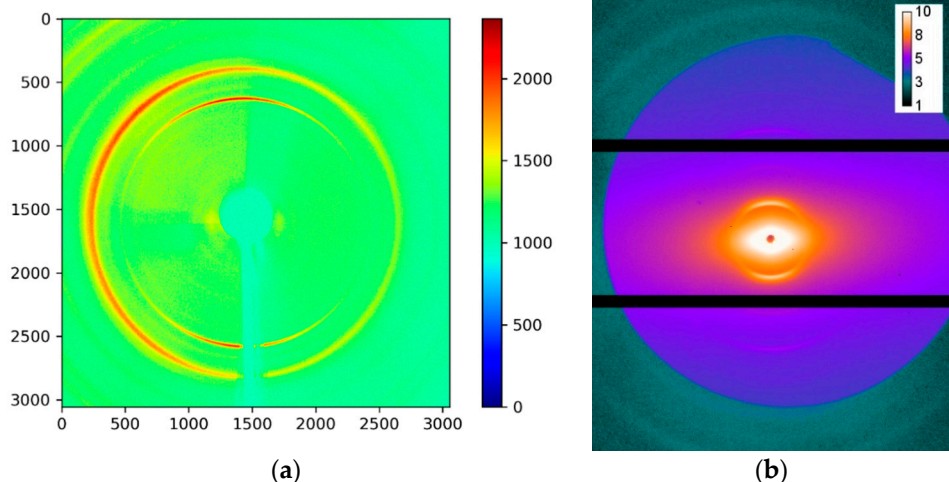

(**a**)                                                                (**b**)

**Figure 2.** 2D (**a**) Wide-Angle X-ray Scattering (WAXS) and (**b**) Small Angle X-ray Scattering (SAXS) patterns, respectively. Colour bars show the intensity in arbitrary units (equivalent counts).

WAXS patterns were collected and interpreted to obtain information about HAp crystalline peaks, as detailed below. SAXS mode was used to primarily to obtain information about collagen. The procedure involved exposure of the WAXS detector for each beam position of the sample, followed by the Image Star detector being translated out of the beam to expose SAXS pattern(s). The procedure was repeated for several loading steps to obtain a raster of SAXS–WAXS exposures to allow the analysis of spatial variation of deformation as a function of applied load.

To accommodate the complex shape of the natural bone, the unique grip setup was designed and fabricated from AA2024-T4 aluminium alloy plate. The loading system allowed applying three-point bending using a titanium M6 bolt with a rounded pin head threaded through the loading frame wall. It was used both to apply the bending force to the bone and to evaluate it performing WAXS from the pin to extract the lattice strain and calculating the stress and force in the bolt by Hooke's law. The experimental setup is illustrated in Figure 3.

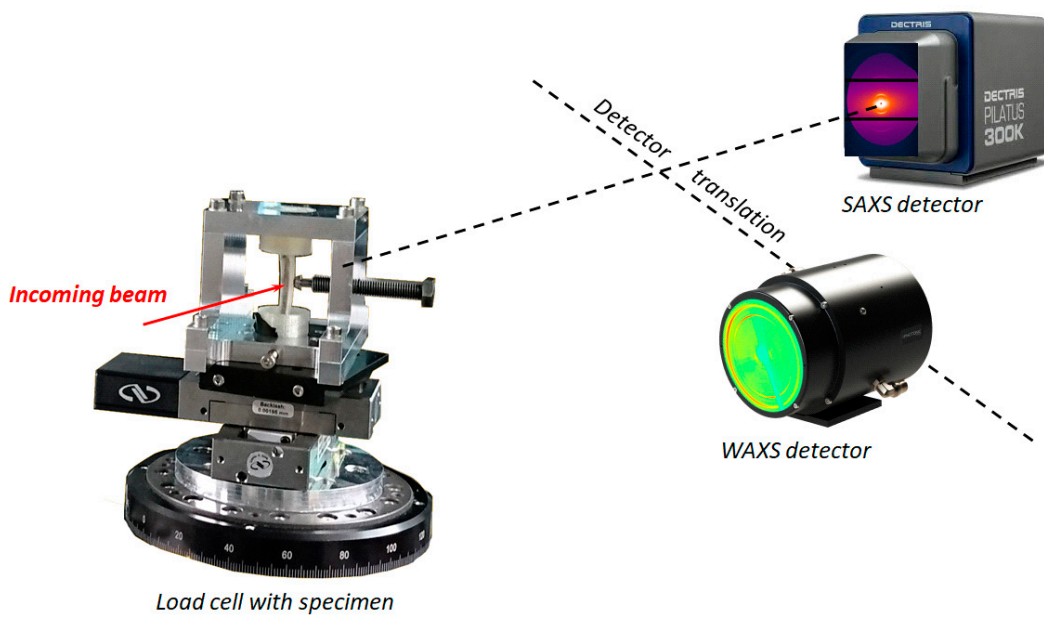

(**a**)

**Figure 3.** *Cont.*

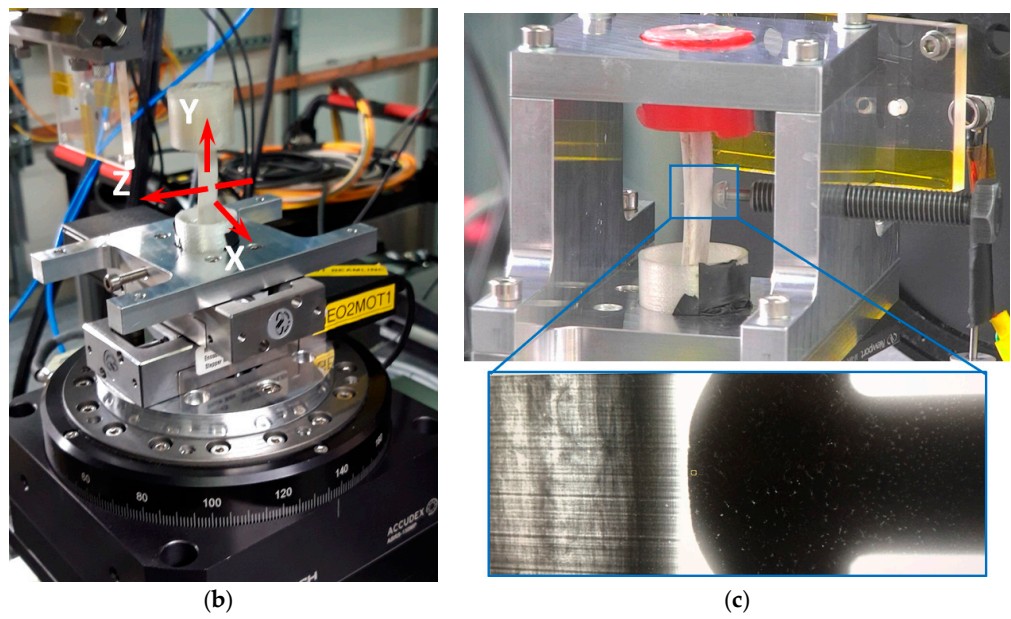

**(b)**  **(c)**

**Figure 3.** (**a**) Experimental synchrotron X-ray diffraction setup with developed load cell: (**b**) specimen position for tomography and (**c**) radiographic image of titanium bolt head used for application during in situ 3-point bending.

## 2.4. Internal Strain Evaluation

An area of 3.2 × 5.6 mm was scanned with the step of 0.2 mm along both $x$ and $y$ directions perpendicular to the incident beam. Overall, 2D wide angle and small angle diffraction patterns were acquired at 448 points for each specimen and load. Each sample was investigated at different loads, and lattice strains were determined using the technique described below. The information from the calibration pattern was processed using the DAWN Science program [31] to obtain precise estimate of the beam centre position in the reference image.

Synchrotron X-ray diffraction is a widely used technique to determine lattice strains within a material, with the crystal planes used as "strain gauges." Lattice straining leads to the distortion of the Debye-Scherrer rings on the detector due to the change in the prevalent $d_{hkl}$ interplanar lattice spacing. This is governed by Bragg's law that can be written as:

$$2d_{hkl}\sin\theta = \lambda,$$
$$q = \frac{2\pi}{d_{hkl}} \tag{1}$$

where $\theta$ is half of the scattering angle, $\lambda$ is the X-ray wavelength and $q$ is reciprocal lattice vector.

Crystal orientation-specific lattice strain ($\epsilon_{hkl}$) can be calculated from the interplanar spacing ($d_{hkl}$) as follows:

$$\epsilon_{hkl} = \frac{d_{hkl} - d_0}{d_0} \tag{2}$$

where $d_0$ denoted the corresponding interplanar spacing in a non-deformed state.

Our recent article [32] proposed an approach that offers an alternative to the conventional azimuthal slicing ("caking") to process 2D WAXS images. This traditional approach to the calculation of lattice microstrains requires a multi-stage process that involves the following stages: calibration, conversion of the 2D pattern into a set of 1D profiles, and fitting for peak centre determination to determine strain values. In this study, each 2D diffraction pattern was separated onto 36 sectors with 5° semi-width in the left (counterclockwise) and right (clockwise) directions for each step in 10°, as shown in Figure 4.

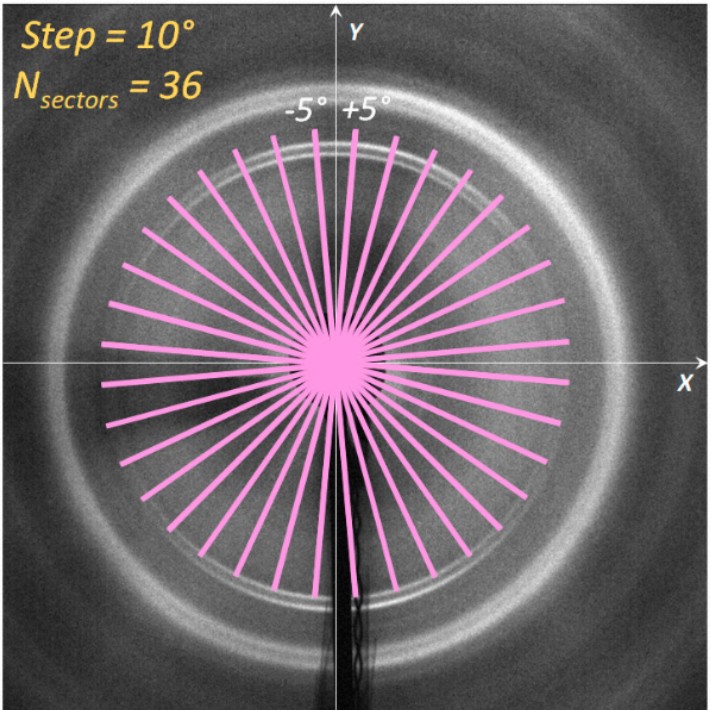

**Figure 4.** Illustration of the 'caking' procedure for the conversion of 2D diffraction pattern to a set of 1D 'radial' intensity distributions.

The centre peak positions were fitted by a double-argument sine function of the azimuthal angle, namely:

$$R_i = a + b \, \sin(2(\phi_i + c)) \tag{3}$$

where $a$, $b$, and $c$ are the mean radius, amplitude, and phase shift, respectively.

Equation (3) describes an ellipse of mean radius $a$ and maximum radius $a + b$ that happens in the direction of maximum strain according to the next formula:

$$\epsilon(\phi) = \frac{\epsilon_1 + \epsilon_2}{2} + \frac{\epsilon_1 - \epsilon_2}{2} \cdot \cos(2\phi) \tag{4}$$

where $\epsilon_1$ and $\epsilon_2$ are principal strains in principal directions that are rotated by angle $\phi$ with respect to Cartesian axes (x, y) considering that $\phi = 0$ is along x.

The evolving lattice strains we evaluated as:

$$\epsilon = \frac{d - d_0}{d_0} = \frac{d}{d_0} - 1 = \frac{a + b}{a_0 + b_0} - 1 \tag{5}$$

where $\epsilon$ is an elastic residual strain, $d = a + b$ is $d$-spacing of a strained sample, and $d_0 = a_0 + b_0$ is the value of $d$-spacing when the sample is nominally strain-free (from calibration image).

The variation of 1D intensity profiles and the changes in the principal peak positions during loading can be observed in Figure 5 [18].

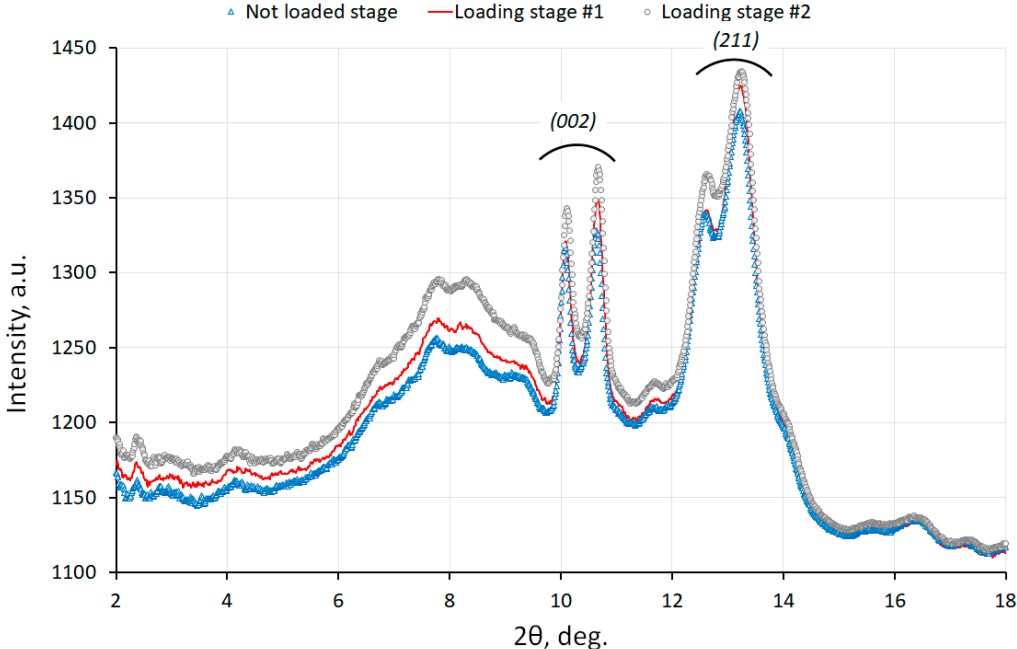

**Figure 5.** Equivalent 1D profiles at various stages of bone loading [18].

## 3. Results

### 3.1. Tomography, Reconstruction and Segmentation

Fast X-ray tomography was carried out for unloaded samples in parallel beam configuration to visualize the bone 3D morphology consisting of distinct component phases and revealing structural inhomogeneity, e.g., in the of tubular internal channels (Haversian canals) and submicron porosity with 5 μm spatial resolution. This stage was carried out for the sample unloaded after cracking. The raw tomographic projections were processed using fast and flexible software package *SAVU* (Diamond Light Source, Harwell, Oxford, UK). An example of a reconstructed tomographic slice is shown in Figure 6.

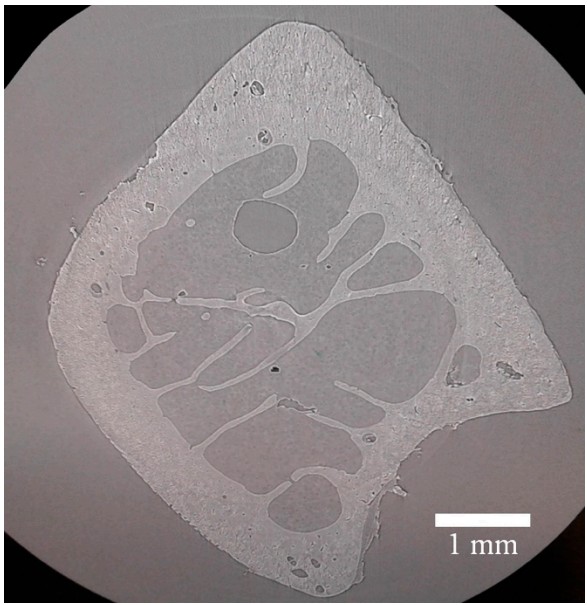

**Figure 6.** An example of tomographic slice a transverse to the bone extent obtained from the reconstruction of radiographic projections.

Segmentation was performed using commercial software (Avizo version 2020.1, ThermoFisher Scientific, Waltham, MA, USA) by applying the following technique. Firstly, the dataset was cropped using Fiji (open-source software for image processing) and then uploaded in Avizo. Segmentation editor was used to identify material types using seeds (selected representative points), and then watershed was used to segment the volume. For 'composite' materials structures (one type of material embedded within the other), interactive thresholding was used. The steps of segmentation processing and details of bone microstructure are illustrated in Figure 7. The processed 3D dataset consisted of $338 \times 314 \times 127$ pixels in 32-bit floating greyscale format corresponding to the physical size of $1685 \times 1565 \times 630$ µm and voxel size of $5 \times 5 \times 5$ µm. The memory used was 51.4 MB.

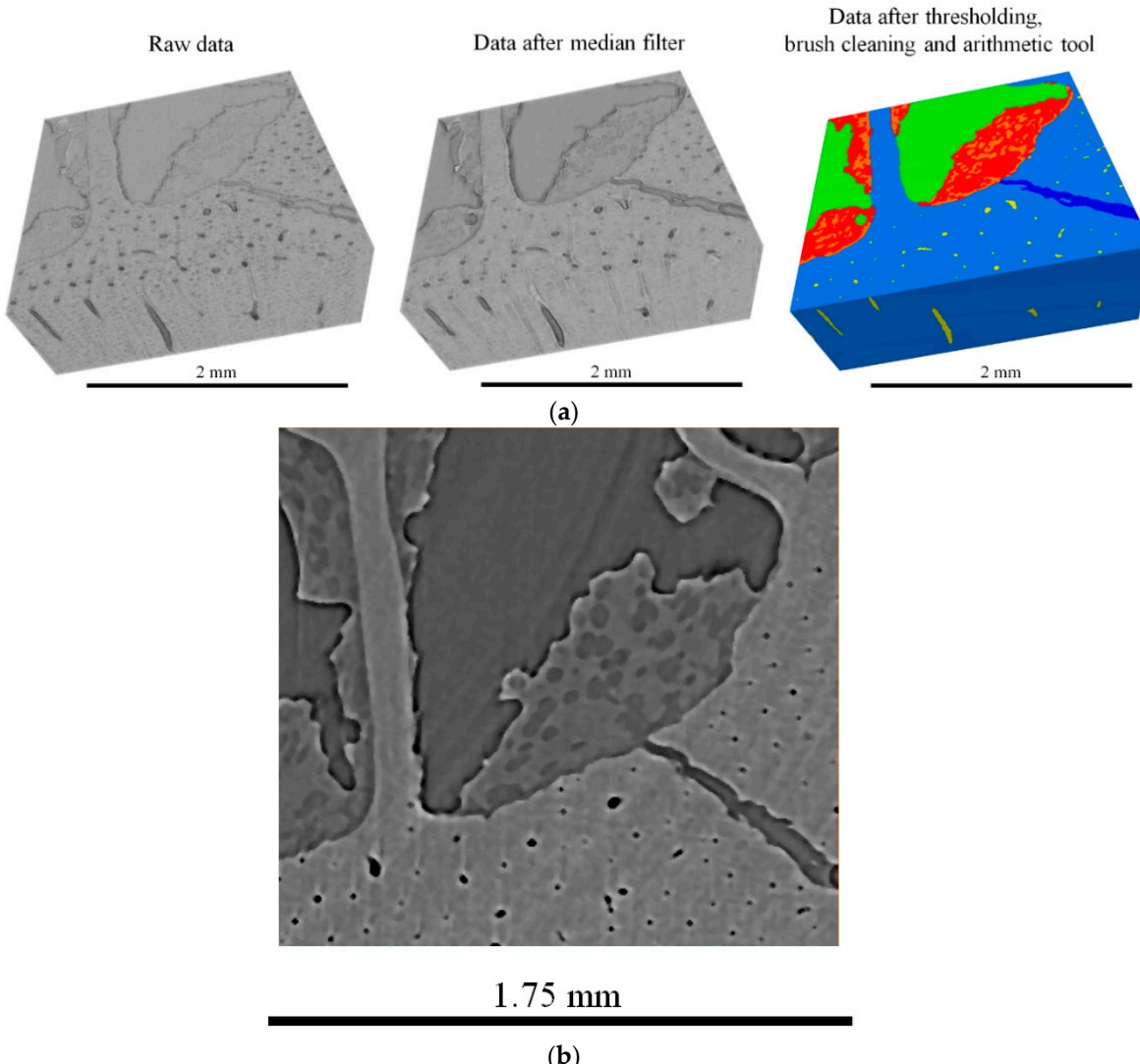

**Figure 7.** (**a**) Segmentation data processing steps; and (**b**) detail showing simultaneous capability of microscale visualization of cells in bone marrow, and cortical bone structure (Haversian canals).

It is apparent from the images shown in Figure 7 that X-ray tomographic reconstruction of the ovine bone cross-section provides an excellent non-destructive tool for bone characterization in terms of geometry (shape, dimensions) and morphology (constitutive cells and structures).

### 3.2. WAXS Analysis

In this article, lattice strains were evaluated through the interpretation of 2D WAXS patterns illustrated in Figure 4 using the approach described in [33]. Processing was performed using bespoke

package written in Python programming language. For the purpose of internal strain evaluation under 3-point bending loading, the vertical direction (along the bone length) and the most prominent (211) diffraction peak of HAp were chosen, as illustrated in Figures 4 and 5, respectively [18]. In the course of processing 1D profiles, there were several types of peaks encountered: single, shouldered, and double peaks, as indicated in Figure 8.

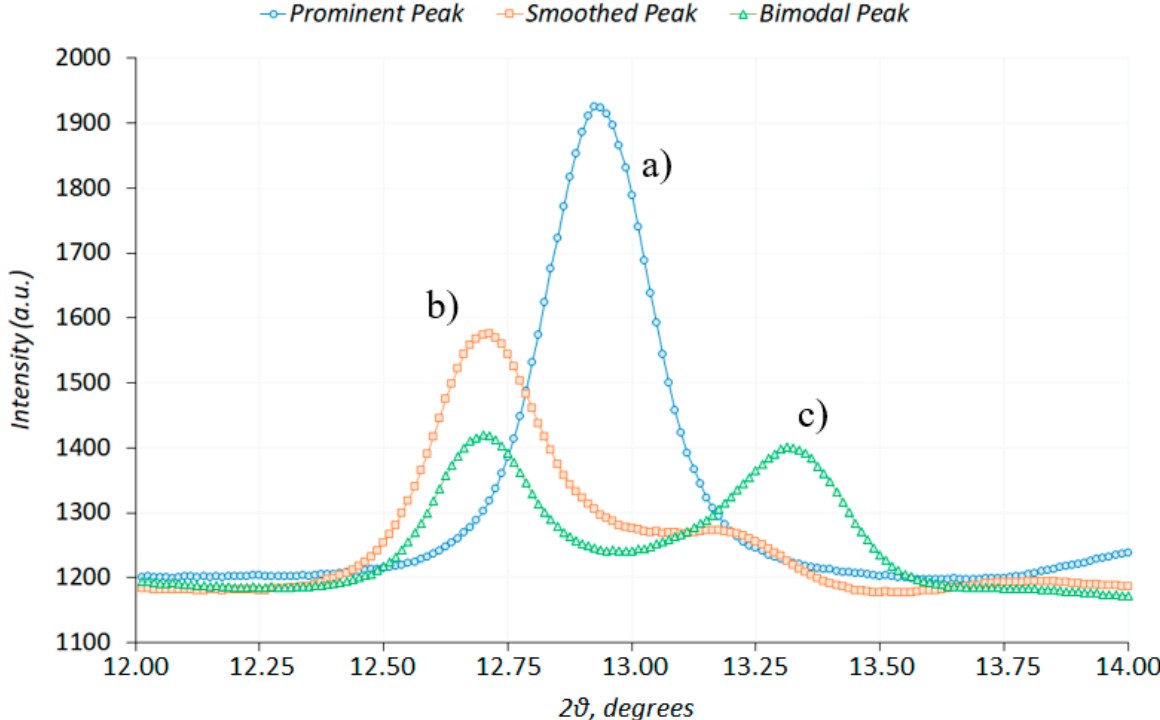

**Figure 8.** The evaluation of diffraction peak centres for (**a**) single; (**b**) shouldered; and (**c**) double peaks [18].

The process of functional fitting of complex peak shapes, such as 'shouldered' peak in Figure 8b, the best approximation needs to be found in terms of the superposition of two known parametric functions, e.g., Gaussian. The process is stable for the case of single or double peaks, but may present significant challenges in the case of 'shouldered' peak that may present an ill-posed problem. Non-linear optimization Python package LMFit was used to address this problem.

A crucial element of challenge and complexity that arises in the study of internal stress-strain state within bone samples arises from the fact that peak position on the WAXS diffraction detector is affected not only by the strain-related changes in the interplanar lattice spacing, but also by the variation in the sample–detector distance. Therefore, in order to extract information about the internal strains alone, the two effects need to be quantified and reliably separated.

The effect of lattice spacing change can be linked to the observed change in the angular position of the ring on 2D diffraction detector by differentiating Bragg's law Equation (1) assuming fixed wavelength $\lambda$, as follows:

$$2\Delta(d_{hkl})\sin\theta + 2d_{hkl}\cos\theta\Delta(\theta) = 0,$$
$$\epsilon_{hkl} = \frac{\Delta(d_{hkl})}{d_{hkl}} = -\frac{\Delta(\theta)}{\tan\theta} = -\frac{\Delta(2\theta)}{2\tan\theta} \approx -\frac{\Delta(2\theta)}{2\theta} \tag{6}$$

where the assumption has been made that the scattering angle $2\theta$ is small. Thus, peak centre displacement on the detector can allow lattice strain to be determined.

However, the complex shape of the sample cross-section can introduce significant 'aberrations' in the position and appearance of Debye–Scherrer rings forming the scattering pattern on the detector,

that manifests itself in the double ring structure seen in Figure 4. The origin of this phenomenon has been previously discussed in an earlier paper devoted to the subject of diffraction strain tomography [34]. In that study, a sample of nearly circular cross-section was considered. It was established that, with the appropriate correction taken for the ring radius determined by the distance to the detector from the centre of gravity of the gauge volume, the peak centre position corresponds to the mean value of the lattice parameter within the entire gauge volume illuminated by the incident beam.

The situation is more complex in the present experiment, as illustrated in Figure 9. It is apparent that that morphology of the bone cross-section illustrated in Figure 6 means that both dense (cortical) and highly porous (trabecular) bone regions are found in the beam path. Moreover, the scattering is dominated by the contributions from the two separate regions of cortical bone that lie at different distances from the detector. Consider similar triangles formed by the incident beam and the two scattered beams. As shown in Figure 9, a larger triangle is formed by the 'front' part of the bone (upstream in the beam) compared to the 'back' part (downstream in the beam), which leads to a larger and smaller radial positions of the peaks on the detector, respectively. For the purposes of this analysis, to the first approximation the small difference in the scattering angle $2\theta$ that may arise due to different strains in the two parts of the bone can be ignored.

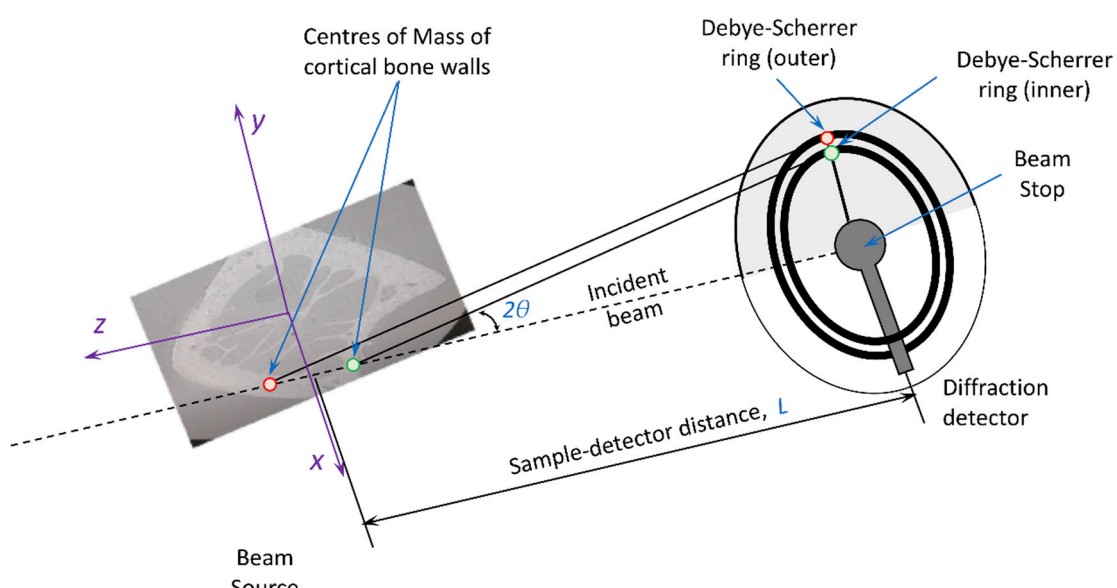

**Figure 9.** Illustration of the effect of bone sample cross-sectional shape on the radial ring position on the 2D diffraction detector.

To evaluate the magnitude of this geometric effect, we determine the apparent change in the Debye–Scherrer ring radius due to the change in the sample-detector distance below. For the purposes of analysis, it is convenient to assume that the 'sample-detector distance' $L$ corresponds to the position of the centre of the rotation stage used in the tomography data collection, which in turn was also used for mounting the calibrants. All coordinates within the bone cross-section can then be referred to this centre. In the beamline 'laboratory' axes, $x$ and $y$ correspond to the horizontal direction to the right of the incident beam and the vertical upwards direction, respectively. The $z$-axis corresponds to the horizontal direction towards the beam source, so that in any sample cross section positive values of $z$ indicate positions upstream from the rotation centre, and negative downstream, as illustrated in Figure 9.

Considering the variation in the position of the gauge volume along $z$, the following relationship can be established:

$$\tan 2\theta = \frac{R}{L} \tag{7}$$

where $2\theta$ is the scattering angle, $R$ is the radius of the Debye–Scherrer ring on the detector for a small sample placed at the centre of rotation that lies the distance $L$ from the detector.

Assuming the small changes in the scattering angle can be ignored, the following relationship applies:

$$\tan 2\theta = \frac{R}{L} = \frac{R + \Delta R}{L + z}, \quad \frac{\Delta R}{R} = \frac{z}{L} \tag{8}$$

Thus, the displacement of the gauge volume upstream of the rotation centre results in the apparent strain:

$$\widetilde{\epsilon}_{hkl} \approx -\frac{\Delta(2\theta)}{2\theta} \approx -\frac{\Delta R}{R} = -\frac{z}{L} \tag{9}$$

where the tilde sign (~) above the strain symbol indicates that it is the apparent, or perceived strain.

The relationships established in Equation (9) confirm that the cross-sectional shape and internal strain within bones samples are coupled, in that they both produce ring radius change on the detector. The analysis of actual internal strain within the sample therefore requires deconvolution of these two effects. We demonstrate below how this can be accomplished with the help of the tomographic imaging information collected in the same experiment.

### 3.3. SAXS Analysis

SAXS pattern analysis was performed by the transformation of 2D pattern to 1D profile, background subtraction, peak centre fitting, and conversion to nominal strain.

To improve diffraction pattern interpretation, the logarithm of SAXS pattern intensity was computed, and conversion to 1D profiles in the 90° (longitudinal with respect to the bone) orientation was performed withing ±15° azimuthal range using Process-Math-Log and Plot Profile functions within the ImageJ program. Raw SAXS data profile, background and the SAXS profile after background subtraction shown in Figure 10. In accordance with reports in the literature, the two prominent peaks present in the SAXS profile were identified with the first and third periods in the structure of collagen fibres that possess the characteristic real space period of 67 nm [35].

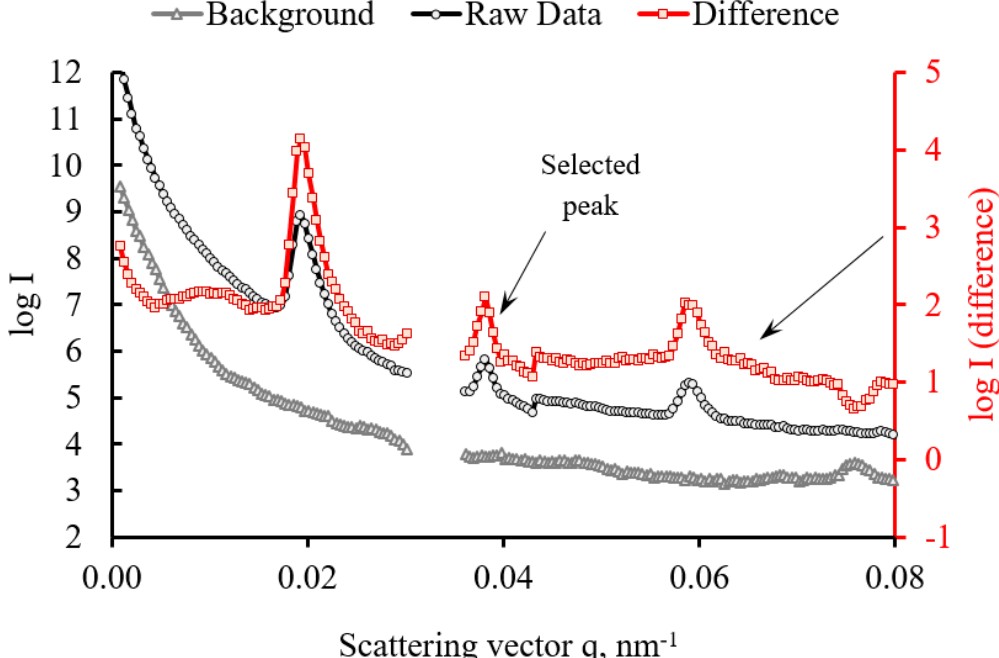

**Figure 10.** Raw SAXS profile (circle markers), background curve (triangle markers) and difference curve (square markers; red curve referred to the right hand scale).

Next, the conversion of SAXS peak centre position to strain was attempted. In spite of the obvious analogy with WAXS interpretation, the mechanism of peak formation and the conversion of its centre position to strain are both different, as noted and discussed previously [36] in the paper devoted to multiple length-scale analysis of deformation in a polymer, and references therein. It was noted that a 'deficit' amounting to a factor of 3 to 4 arises when the nominal SAXS strain derived from relative peak displacement is compared to the equivalent macroscopic strain.

Each peak was fitted by a Gaussian function, and the centre position was used to calculate the nominal strain using the formula $\epsilon = -\frac{p-p_0}{p_0}$, where $p$ is the peak centre position in pixels (considering the small scattering angle), and $p_0$ was the average peak centre position.

## 4. Discussion

We use the above Formulae (7)–(9) to convert the radial position of the peak on the detector into the equivalent position within the sample cross-section, as illustrated in Figure 11. It is apparent that the peak displacement on the detector is dominated by the sample shape. In fact, the two sets of dots present in the upper and lower parts of the cortical bone as shown in Figure 11 were derived from the positions of individual peaks within the doublet illustrated in Figure 8. This is a consequence of the inhomogeneous bone structure (cortical and trabecular bone), the scattered beam intensity dominated by the contributions from the high density cortical regions. However, it also becomes clear that the apparent small deviations from cortical wall centre of mass positions along the beam are related to elastic strain effect. Moreover, we are looking at strain along $y$-axis according to the Figure 3b.

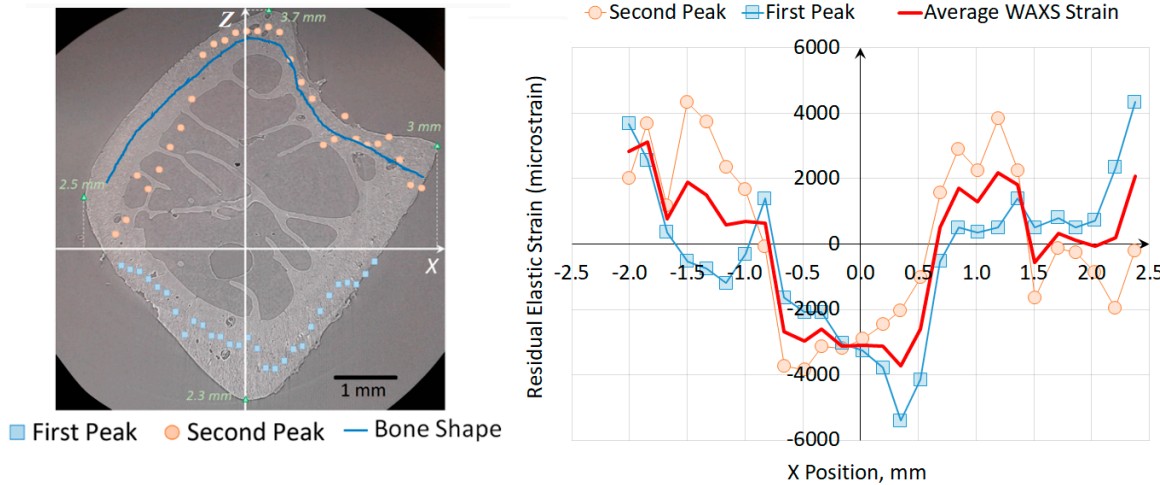

**Figure 11.** Illustration of the influence of bone morphology on the apparent peak position on the detector, and the extraction of inherent lattice strain by incorporating bone shape correction.

Thus, the observed changes in the radii of Debye–Scherrer rings consists of two effects: (i) the variation of the gauge volume centre of mass to detector distance due to the complex sample shape, and (ii) the strain-related change in the lattice interplanar distance $\Delta(d_{hkl})$ that affects the scattering angle through Bragg's law.

This provides direct experimental and interpretational motivation for the incorporation of this two-pronged approach in the analysis algorithms to be developed and adopted for DIAD (Dual Imaging and Diffraction) instrument and paradigm.

In Figure 12, the results of strain analysis by synchrotron X-ray diffraction in the unloaded ovine rib bone are plotted as WAXS-based strain together with the SAXS strains multiplied by the multiplication factor of 4. SAXS data points shown in the plot were subjected to filtering on the basis of error associated with peak fitting. The combined results reveal a consistent profile that shows a clear correlation between the two plots, both displaying a characteristic 'bent bar' shape that arises due to

bending of a beam beyond its elastic limit [37], as indicated by the grey solid line in the plot as a guide to eye.

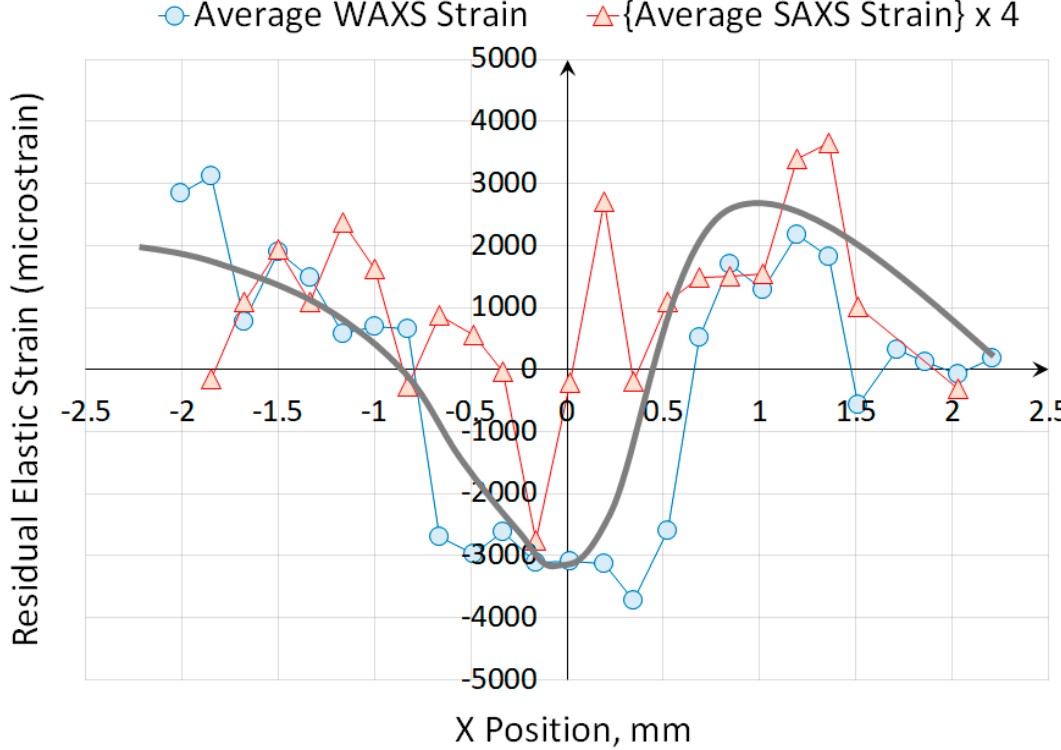

**Figure 12.** The comparison of strains obtained from the interpretation of WAXS and SAXS patterns. The continuous curve represents a guide to the eye.

Inelastic bending leads to residual stress (and residual elastic strain) by means of imposing permanent 'frozen in' strain (or eigenstrain) that generates misfit upon unloading, and gives rise to residual tension observed on the side that underwent plastic compression, and vice versa. To the best of the authors' knowledge, there has not been an attempt reported of building an analogy between this purely mechanical phenomenon, and the more complex process of bone growth into a permanently bent shape characteristic of a rib. In this context it may be salutary to consider the process and its consequences in line with the *mechanostat* hypothesis. The 'shorter' side of the bone experiences slower growth that may be associated with the deposition of greater amounts of HAp and local compressive stresses. Upon completion of bone growth into a curved shape, internal stress imbalance leads to straightening (unbending) that 'flips' the distribution and leads to the profile observed in Figure 12. Thus, growth-initiated permanent internal strains determine the residual stress distribution in a way that is similar to the effect of plasticity-induced strains in a mechanically bent bar. Further consideration of the plots in Figure 12 reveal positive correlation between WAXS-related strain that is linked to the residual stress within HAp nanocrystals, on the one hand, and the SAXS-related strain that is connected with the strain within collagen fibres, on the other.

As can be seen from the plots in Figure 12, in the central part of the bone (the region corresponding to range $0 < x < 0.5$ mm position), significant difference is observed between residual strains in the organic part (collagen) evaluated using SAXS, and the strain in the mineral part (HAp) obtained from WAXS. In this region the effect of 'plasticity' is negligible, and the collagen is found to carry residual tension, while HAp is placed in compression.

This is consistent with the *OsteodynE* hypothesis formulated above, namely, that the organic and mineral components of mammalian bone exert internal stresses upon each other that are purposely built into the tissue to optimise its mechanical performance: the fibrous organic part is held in tension,

compressing the mineral part that is more brittle, but capable of carrying large loads in compression without failure.

## 5. Conclusions

The *OsteodynE* hypothesis put forward in this article presently remains a hypothesis, as its profound and reliable validation requires significant further work. Nevertheless, it is worth in the discussion to connect and compare it with the current opinion that can be found in the literature on the subject. The elucidation of the hierarchical structure of bone, and the way that the arrangements at each length scale exert influence of the overall mechanical properties have been addressed in a recent overview article [37]. In particular, it was stated therein that "plasticity (permanent deformation) in bone results from concurrent multiple deformation mechanisms that are active at all hierarchical levels". A role of particular importance is played by individual collagen molecules that undergo stretching and unwinding involving hydrogen bond breaking and re-making, e.g., through slip and stick between fragments of tropocollagen molecules. It is worth noting that this crucial 'healing' capability is a property of young bone, since the increasing cross-link density in older tissue impedes this mechanism of dissipate energy without failure. Additionally, it is worth noting the way in which collagen fibrils are intrinsically mineralized with hydroxyapatite to confer increased stiffness on the bone tissue. Recoverable bonds are also active at coarser levels, with fibrils held together with weaker protein "glue" that debonds at 2- to 10-times lower force than that required to break the macromolecular chains. These bonds can re-form in the presence of body fluid in a manner similar to hydrogen bonds at the finer molecular scale.

The contribution we seek to make in the present study to the understanding of bone strength and toughness is to focus attention on the internal load transfer and stress balance between different regions within the bone at the millimetre scale, and between organic and inorganic parts of the bone tissue at the micro-scale. In the latter respect in particular, the collagen network maintains tension against HAp micro- and nanocrystals that are held in compression, improving the ability of the entire "composite" to accommodate tensile deformation.

**Author Contributions:** Conceptualization, A.M.K. and A.I.S.; methodology, I.P.D., A.M.K., A.I.S. and T.M.; software, E.S.S. and C.B.; validation, E.S.S., A.M.K. and A.I.S.; formal analysis, E.S.S.; investigation, J.C., C.B., Z.W. and T.M.; writing—original draft preparation, E.S.S., A.I.S. and A.M.K.; writing—review and editing, A.M.K., A.I.S. and I.P.D.; visualization, E.S.S., J.C. and C.B.; supervision, A.M.K.; project administration, I.P.D. All authors have read and agreed to the published version of the manuscript.

**Funding:** A.M.K. acknowledges the support for this research by the Royal Society of London through grant IEC/R2/170223.

**Acknowledgments:** The authors acknowledge the access to Test Beamline B16 at Diamond Light Source, UK, provided through beam time allocation MT21419-1.

**Conflicts of Interest:** The authors declare no conflict of interest.

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
