# Peer review of "Ovine Bone Morphology and Deformation Analysis Using Synchrotron X-ray Imaging and Scattering"

_qubs, doi:10.3390/qubs4030029_

Round 1

Reviewer 1 Report

The manuscript, “Ovine Bone Morphology and Deformation Analysis Using Synchrotron X-Ray Imaging and Scattering” by Eugene S. Statnik, et al. reports internal residual strain distribution of hydroxyapatite and collagen fibers in an ovine bone using synchrotron x-ray imaging and scattering techniques.  This work could contribute to understand internal bone structure more deeply.  I would like to recommend this manuscript for publication in Quantum Beam Science after revisions.  Comments and questions to authors should be addressed;

- Authors introduced a specially designed in-situ load cell using 3 point bending.  There is also descriptions regarding in-situ loading throughout this manuscript.  However, all results in this manuscripts are from unloaded sample.  I am curious why there is no data depending on loading steps.  If authors can’t show data on various loading, I suggest remove descriptions regarding in-situ loading including the load cell.  If datasets from in-situ loading are available, authors need to extract the stress and the force at each step.

- In addition to the question above, what is the origin of residual elastic strain shown in Fig. 10 and 12?  Is it from the sample preparation process?

- Authors used white x-ray beam from Diamond Light Source for tomography.  I wonder if the white x-ray beam caused any damage on the sample and how authors checked this issue.

- Is there any reason why authors chose different x-ray beam size for WAXS and SAXS?

- There is a circular shadow in the image (Fig 2(b)).  What is the origin of this shadow?  I don’t think a set of slits makes this shadow.

- How do authors extract 1D line profiles in Fig. 4 from 2D datasets?  It is not clearly mentioned.

- The sentence, ‘For the two materials inside other (orange and yellow) segmentation editor was used in order to perform interactive thresholding inside the material already segmented’ in line 270 through 272, is not clear.  I guess it is related to Avizo software.

- Is there any particular reason to choose (211) Bragg peak of Hap instead of (002) Bragg peak for internal strain evaluation?

- Authors assume scattering angle, 2θ, is small for Eq. 6.  However, in reality, 2θ is not small.

- I am confused by axes which authors used.  Authors mentioned the laboratory axes point outboard, up, and upstream (x, y, z, respectively).  Direction of x and y is not clear in Fig. 9 (I think y axis has to be modified).  Y axis in Fig. 10 does not make sense (I think y has to be labeled as z).  It makes readers hard to understand.

- What is the meaning of First and Second peak which are labeled in Fig. 10?  Are they related to Upper WAXS and lower WAXS?  Still these terms are not clear.

- What is the meaning of 'longitudinal w.r.t. the bone'?  Do authors mean along the length of the bone?

- In Fig 3, there is no figure labeled as (a).

- I think there is a typo at the end of line 96.

- It might be better to use 2θ (degrees) instead of Profile Line (pixels) for Fig. 8 because pixel number has no meaning.

Author Response

Authors would like to thank the reviewer for the detailed comments and suggestions for the manuscript. We believe that the comments have identified important areas that required improvement. After completion of the suggested edits, the revised manuscript has benefitted from an improvement in the overall presentation and clarity. Below, you will find a point by point description of how each comment was addressed in the manuscript. Please see the attachment.

Reviewer 2 Report

The authors presented a modern approach to medical imaging aiming the investigation of structure and deformation of ovine bone. The presented considerations are extremely interesting and have great scientific significance, however, the level of this article is not satisfactory.
First: The correct structure of the article was not preserved, i.e. the introduction is very general, the results are described too extensively, while the discussion was very succinct.
Second: It was confirmed, that formalin fixation makes significant changes in material characteristics of the bone, especially in the stiffness of bones, which undermines the conclusions drawn.
Third: Averaging the measurement results and limiting them to a single scan does not allow the bone mechanics to be determined taking into account its microstructure. The analysis carried out should be extended to several measurement areas in one bone.

Author Response

(The authors gave the same response as above.)

Round 2

Reviewer 1 Report

I appreciate all of authors’ efforts to improve this manuscript.  This revised manuscript becomes more clear than the previous one.  I would like to recommend this manuscript for publication in Quantum Beam Science after minor revisions.  Comments to authors are in the followings;

- In Fig. 11, both the image and the graph have their own legends.  However, it might be better if legends on the left and the right are matched.  I think ‘First peak’ and ‘Upper WAXS’ have same meaning which is peaks from the bone wall at Z<0 whereas ‘Second Peak’ and ‘Lower WAXS’ from Z>0.  It will be better to keep ‘Upper WAXS’ and ‘Lower WAXS’ since these terms are explained in this manuscript.  I also suggest to change the color of graphs to match the one in the image.  This will help readers understand Fig. 11 more clearly.

- I found one typo in line 274, ‘ow’ instead of ‘of’.

Author Response

Authors would like to thank the reviewer for the detailed comments and suggestions for the manuscript. We believe that the comments have identified important areas which required improvement. After completion of the suggested edits, the revised manuscript has benefitted from an improvement in the overall presentation and clarity. Below, you will find a point by point description of how each comment was addressed in the manuscript.

Reviewer 2 Report

Thank you for your response. Generally I accept your corrections.

Author Response

The authors would like to thank the reviewer for the detailed comments and suggestions for the manuscript.